# Rapid Heating of Mold: Effect of Uneven Filling Temperature on Part Morphology and Molecular Orientation

**Sara Liparoti ***, **Daniele Sofia** and **Roberto Pantani ***

Department of Industrial Engineering, University of Salerno, Via Giovanni Paolo II 132, 84084 Fisciano, SA, Italy
* Correspondence: sliparoti@unisa.it (S.L.); rpantani@unisa.it (R.P.)

**Abstract:** Mold temperature is the key parameter in determining the morphology of molded parts. Uneven temperature distribution could induce significant effects on part performances. In such cases, uneven temperature is induced to analyze the morphology developed in the molded specimens. The technology used for controlling mold temperature during the process is crucial to maintain the short processing time. This paper proposed a strategy for controlling mold temperature during the process, avoiding a significant increase in processing time. A thin electrical heater is designed and adapted below the cavity surface, allowing for the increase of the cavity surface temperature soon after the mold closure, and the fast decrease of the mold temperature soon after the filling. The effect of several heating powers and heating durations on the molecular orientation was analyzed and exploited considering the temperature and flow field realized during the process.

**Keywords:** injection molding; process simulation; molecular orientation

## 1. Introduction

Injection molding is one of the processes encountering high interest from the industrial community due to its high productivity, high automation, and high geometrical accuracy, which make the process suitable for a wide range of applications [1]. The process is mainly composed of three stages: the filling of a molten polymer into the cavity, the additional feeding of melt to compensate for the shrinkage due to solidification (namely the packing), and the cooling down to the ejection temperature. Injection molding is generally conducted with a low cavity temperature to make the last stage as fast as possible and increase the process productivity. This approach may suffer due to the large temperature difference between the cavity and the melt. At the first contact of the melt with the cavity surface, polymer solidification begins almost instantaneously, inducing the formation of a frozen layer on the part surface. This causes difficulties in cavity filling, especially for cavities with small thicknesses where cooling is faster, surface finishing is poor, and there is poor accuracy in replicating micro- and nano-metrical features [2–6]. The increase of cavity temperature during the filling and packing has proven to positively affect several part and process characteristics. For instance, the molded part's surface appearance, strength, and shape accuracy can be significantly enhanced by the increase of the cavity temperature. Moreover, the required injection pressure and clamping force of the molding machine significantly decrease when increasing the cavity temperature, leading to a significant energy saving [2,7–10]. However, the increase of the cavity temperature may induce an increase in the processing time, particularly the cooling stage, with a consequent decrease in the productivity of the injection molding process. This adverse effect limits the use of this strategy in the industrial field, where the preferred approach is to set a low mold temperature despite the molded parts' poorer qualities.

Recently, the application of injection molding in the biomedical field has made it mandatory that the molded parts show characteristics such as high mechanical strength, close dimensional tolerances, high shape accuracy, low residual stress, good appearance,

and excellent surface texture [5,11–13]. Therefore, great efforts have been devoted to designing strategies allowing rapid cavity heating, assuring a fast cavity increase/decrease rate. In other words, the strategies need to maintain the high cavity temperature during the filling, in such cases also during the packing, and maintain the low temperature during the cooling to maintain high productivity.

Several approaches have been proposed in the literature to modulate the cavity temperature during the process, such as introduction of radiation sources, induction coils, and proximity heating. The induction heating is realized by introducing a cavity insert wound with induction coils, which allows temperature modulation [14]. The radiation heating is based on the introduction of a movable insert that can be heated by an infrared source [8]. The proximity heating mainly consists in the flowing of a high-frequency current between the two plates of the mold, and this induces a rapid mold heating, while the cooling channels allow for a mold temperature reduction after the current deactivation [15]. Despite the great impact on the part characteristics, these technologies imply a heating/cooling rate of about 10 K/s, not so fast as to guarantee high productivity, and significant design and tool costs [15–18].

Recently, methods based on introducing flat electrical heaters below the cavity surface have been proposed [19]. The advantage of these methods is the low cost and easy maintenance of the electrical heaters, which could be applied in the industrial field.

In this work, a thin electrical heater film is designed and tested to maintain the surface temperature relatively high during the filling stage. The device is composed of a copper pattern applied on a PET layer, which acts as a thermal and electrical insulator. A fast temperature increase/decrease rate will be assured by the heater design, with the purpose of guaranteeing efficient cooling at the packing end. Injection molding tests were performed on polypropylene (iPP)to explore the heating power's effect on molecular orientation [20,21]. Results were examined on the basis of the numerical simulations of the process.

## 2. Materials and Methods

Montell (now Basell) supplied the polypropylene grade adopted in this work. Its commercial name is T30G (non-nucleated, Mw = 376,000, Mw/Mn = 6.7, tacticity = 87.6%).

A 70-ton Negri-Bossi reciprocating screw injection molding machine was used for the experiments. A rectangular cavity was adopted, with a length L = 70 mm, width W = 20 mm, and thicknesses S = 1 mm. Figure 1 shows the geometry of the cavity used for the injection molding tests.

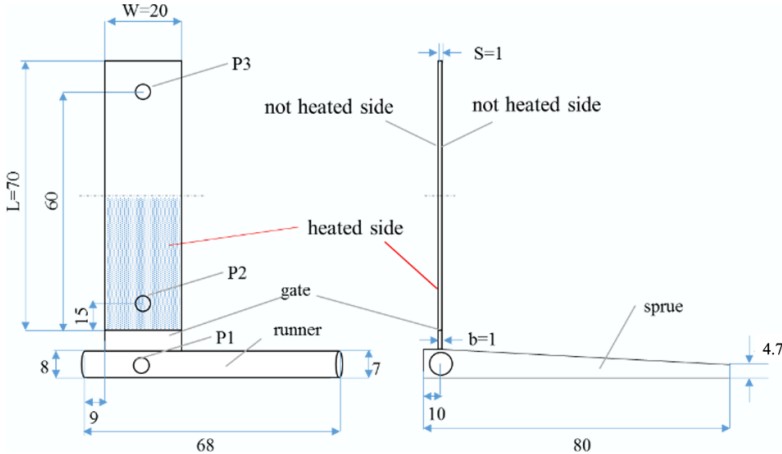

**Figure 1.** Cavity sketch and dimensions.

The molding machine and the mold were equipped with four piezoelectric transducers: one in the injection chamber (P0), one just before the gate (P1), and two in the cavity (P2 and P3), located in the non-moving part of the mold (15 and 60 mm downstream from

the gate position). Moreover, two temperature sensors were located inside the cavity, in position P2, one above the heater, and another one below the heater.

The injection molding processing conditions were: 300 bar as the packing pressure, 1.5 s as the packing time, 2.3 s as the injection time, 220 °C as the melt temperature, 23 °C as the mold temperature, and 1 mm as the cavity and gate thickness (the gate acted simply as a small increase of the cavity length). Injection molding parameters were selected accounting for the main characteristics (melting temperature, rheology, and crystallization temperature [22,23]) of the iPP adopted in this work. Temperature on the cavity surface was modulated by a purpose-built flexible heater, described below. In particular, two heating powers were adopted (they affect the temperature achieved on the cavity surface before the melt contacts the cavity), 40 and 68 W. Conventional injection molding tests, with a constant temperature for the whole mold, were also conducted for comparison.

To analyze the morphology distribution along flow and thickness directions, thin slices, cut in positions P2 and P3 along the flow thickness plane, were analyzed by micrographs in cross-polarized optical light (Olympus BX51, Hamburg, Germany).

## 3. Flexible Heater Design

A copper foil (20 µm thickness) layered on a polyethylene terephthalate film (PET, 20 µm thickness) was used to create the pattern for the electrical resistance. The pattern has a whole length of 835 mm, and a 1 mm width (0.2 mm distance among patterns). This design allows achieving a 2 ohm whole resistance and allows a homogeneous heating of the cavity surface. An insulating PET layer (140 µm thickness) was located between the resistance and the mold to reduce heat dissipation toward the mold. This device was applied on a side of the cavity, as shown in Figure 1, and that side was called "heated". The design of the heater is shown in Figure 2a, and Figure 2b shows the arrangement of the heater in the mold cavity.

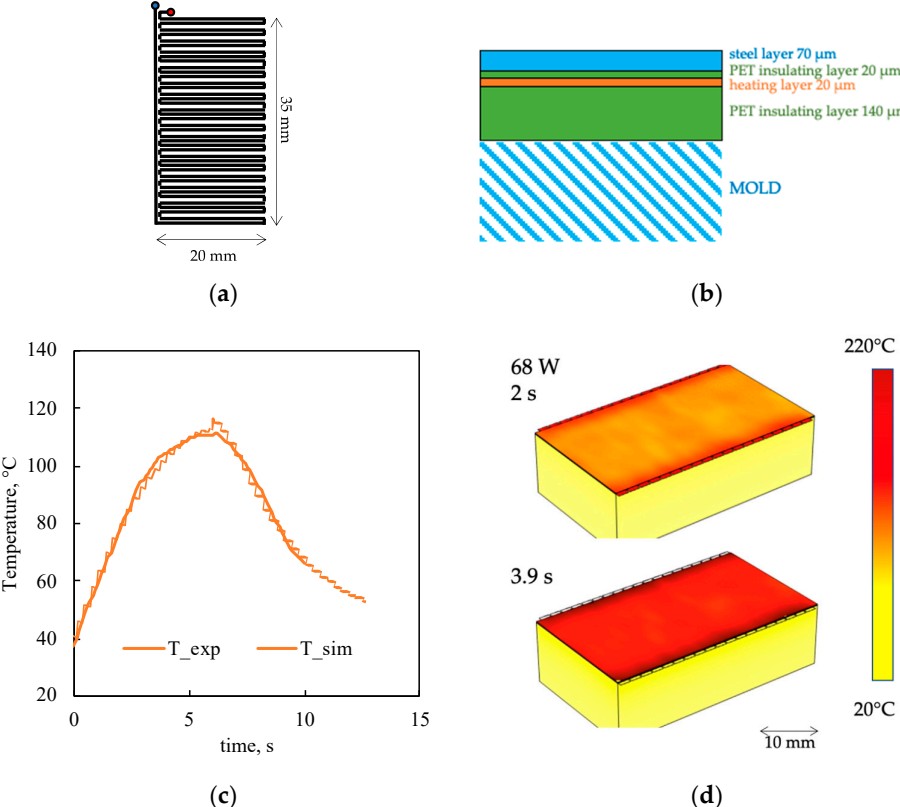

**Figure 2.** (**a**) Pattern of the copper heater. (**b**) Arrangement of the heater in the mold cavity. (**c**) Temperature evolution measured on the cavity, up the steel layer, for a 68 W heating power. (**d**) Simulation of the heating device's activation for the case with a 68 W heating power (2 and 3.9 s).

The heater design (distances and width among paths) was optimized through simulations. The pattern design was implemented in Comsol Multiphysics (ver. 5.6), and the heat transfer model was used. A generation term was adopted to simulate the activation. The insulating layers made of PET and the mold blocks (10 mm thickness, equal to the thickness of the cavity insert) were also included in the domain 3D geometry (number of vertex elements 480, boundary elements 83,391, elements 83,391). The properties used for each layer are reported in Table 1. A temperature of 20 °C was adopted as the initial condition in the whole domain, and continuity of the heat flux was assumed among the layers.

**Table 1.** Properties adopted to model the heating device.

| Parameters | Values |
|---|---|
| PET specific heat capacity (J (kg K)$^{-1}$) | 1030 |
| Cu specific heat capacity (J (kg K)$^{-1}$) | 417 |
| Steel specific heat capacity (J (kg K)$^{-1}$) | 420 |
| PET density (kg m$^3$) | 1400 |
| Cu density (kg m$^3$) | 9000 |
| Steel density (kg m$^3$) | 7612 |
| PET thermal conductivity (W (m K)$^{-1}$) | 0.18 |
| Cu thermal conductivity (W (m K)$^{-1}$) | 395 |
| Steel thermal conductivity (W (m K)$^{-1}$) | 45 |
| Cu electrical conductivity ($\Omega$ m) | $1.6 \times 10^{-8}$ |

Once the pattern was determined through simulations, the copper chip was produced with a laser beam made by using a $CO_2$ laser (emission wavelength at 10.6 μm) with a nominal power of 40 W. The copper layer was fixed onto an x-y plane and moved by stepping motors. A 100 mm s$^{-1}$ scanning velocity and a 100 μm spot diameter were adopted. The laser power density (P) was equal to $2.5 \times 10^4$ W cm$^{-2}$.

Figure 2c shows the temperature evolution recorded on the steel layer (see Figure 2b) during a heating cycle conducted with a 68 W heating power, without injecting the polymer. The heater induced an increase of temperature with a rate of 20 °C/s, and such a rate depends on the heating power adopted. At the heater deactivation, the temperature decrease rate was as fast as the temperature increase rate, thanks to the thin device adopted for controlling the temperature. Figure 2c also shows the simulation of the heating stage, and the consistency of the two evolutions confirms the proper design of the device. Figure 2d shows that the temperature distribution over the surface, in particular on the steel layer, was uniform, regardless of the activation time.

## 4. Modeling

Transport equations, presented as Equations (1)–(3), were used to describe the temperature and the flow field during the process:

$$\rho \frac{\partial \mathbf{u}}{\partial t} + \rho(\mathbf{u} \cdot \nabla)\mathbf{u} = -\nabla p - \nabla \cdot \boldsymbol{\tau} \tag{1}$$

$$\frac{\partial \rho}{\partial t} + \nabla \cdot (\rho \mathbf{u}) = 0 \tag{2}$$

$$\rho c_p \left( \frac{\partial T}{\partial t} + \mathbf{u} \cdot \nabla T \right) - \nabla \cdot k \nabla T = \boldsymbol{\tau} : \nabla \boldsymbol{u} \tag{3}$$

$$\boldsymbol{\tau} = -\eta \left( \nabla \mathbf{u} + (\nabla \mathbf{u})^T \right) + \frac{2}{3}\eta(\nabla \cdot \mathbf{u})\mathbf{I} \tag{4}$$

where **u** is the velocity vector, p is the pressure, T is the temperature, **I** is the unit tensor, $\eta$ is the viscosity, $\rho$ is the density, k is the thermal conductivity, and $c_p$ is the specific heat capacity (see Table 2).

**Table 2.** Properties adopted to model the process.

| Properties | Values |
|---|---|
| Viscosity (Pa s) | $\eta_{air} = -8.4 \times 10^{-7} + 8.4 \times 10^{-8}T - 7.1 \times 10^{-11}$ $T^2 + 4.6 \times 10^{-14} T^3 - 1.1 \times 10^{-17} T^4$ |
| Density (g cm$^{-3}$) | $\rho_{iPP} = 0.900$ Ideal gas behavior for air |
| Thermal conductivity (W (m K)$^{-1}$) | $k_{iPP} = 0.2$ $k_{air} = -2.27 \times 10^{-2} + 1.15 \times 10^{-4} T - 7.90 \times 10^{-8} T^2$ $+ 4.12 \times 10^{-8} T^3 - 7.45 \times 10^{-15} T^4$ |
| Specific heat capacity (J (kg K)$^{-1}$) | $Cp_{iPP} = 2000$ $Cp_{air} = 1047.64 - 0.37\, T + 9.45 \times 10^{-4}\, T^2 - 6.02 \times 10^{-7}\, T^3$ $+ 1.29 \times 10^{-10}\, T^4$ |

In this case, the heat generation due to crystallization was not considered.

The viscosity of the polymer is given by the cross-WLF function (Equation (5)):

$$\eta = \frac{\eta_0}{1 + \left(\frac{\eta_0 \dot{\gamma}}{\tau}\right)^{1-n}} \tag{5}$$

where $\dot{\gamma}$ is the shear rate, evaluated as $\dot{\gamma} = \sqrt{\frac{1}{2}\dot{\gamma} : \dot{\gamma}}$, $\eta_0$ is the viscosity at low shear rates, which depends on the temperature (Equation (6)), $\tau = 10^4$ Pa, and $n = 0.34$, and these values were experimentally determined [24].

$$\eta_0(P, T) = D1 \times 10^{\frac{-A1(T-T^*-D3P)}{A2+T-A3}} \tag{6}$$

It was found that $\eta_0$ abruptly increased up to $10^6$ Pa s below 360 K, and this can be considered a solidification criterium for the process and compensate for the crystallization effect on viscosity. This results in a no-flow temperature, often adopted in the simulation of injection molding.

The Cahn–Hilliard equation (Equation (7)) was used to track the diffuse interface separating air and the polymer (region where the dimensionless phase field variable, $\phi$, goes from $-1$ to 1):

$$\frac{\partial \phi}{\partial t} + \mathbf{u} \cdot \nabla \phi = \nabla \cdot M \nabla P_c \tag{7}$$

where M is a diffusion coefficient ($M = \chi \frac{3\varepsilon\sigma}{\sqrt{8}}$, where the interface thickness, $\varepsilon$, depends on the local mesh size, $\sigma$ is the surface tension (23 mN m$^{-1}$ for polypropylene on a steel surface [20]), and $\chi$ is the mobility tuning parameter equal to 1). $P_c$ represents the chemical potential:

$$P_c = -\nabla \cdot \varepsilon^2 \nabla \phi + \left(\phi^2 - 1\right)\phi \tag{8}$$

The molecular orientation was described through a non-linear formulation of a Maxwell model, in which elastic dumbbells represent polymer macromolecules. **A** is a tensor representing the deformation of the dumbbells with respect to the equilibrium state, the evolution of which is given in Equation (9):

$$\frac{D}{Dt}\mathbf{A} - (\nabla \mathbf{u})^T \cdot \mathbf{A} - \mathbf{A} \cdot (\nabla \mathbf{u}) = -\frac{1}{\lambda}\mathbf{A} + (\nabla \mathbf{u})^T + (\nabla \mathbf{u}) \tag{9}$$

$\lambda$ is the dominant relaxation time, given by Equation (10):

$$\lambda = \frac{c\,\alpha(T, P)}{1 + (a\Delta)^b} \tag{10}$$

*b* and *c* are material parameters, also presented in Table 3, and the shift factor $\alpha = 10^{\frac{-2.5(T-T^*-D3P)}{574+T-A3}}$ (*D*3 and *A*3 are presented in Table 2, for both viscosity and relaxation time).

**Table 3.** Constants adopted in Equations (1)–(10).

| Parameters | Values |
|---|---|
| A3 | 503 K |
| A1 | 1.74 |
| D1 | 7664 Pa s |
| D3 | $1.8 \times 10^{-6}$ m s$^2$ K kg$^{-1}$ |
| T * | 503 K |
| c | 15 s |
| b | 2.2 |

The molecular orientation, $\Delta$, is given by the difference between the two main eigenvalues of the tensor **A** [20].

The suite Comsol Multiphysics (ver. 5.5) equipped with heat transfer and microfluidic modules has been used to build and solve the mathematical model, boundary, and initial conditions. The whole domain was then divided into finite elements, building an unstructured mesh consisting of Lagrange quadratic elements (number of elements: 2352) over the entire domain of the cavity and of linear elements over the boundaries (number of elements: 612). Independency of the numerical solution with respect to the mesh sizes was also conducted to select the proper number of elements that guarantee a fast calculation time and reliable results at the same time. Four segregate solvers have been used for solving the phase field, the flow field, the heat transfer, and the purpose-built model for molecular orientation. The degrees of freedom were 2655. A 2D geometry has been adopted (60 mm length, 1 mm width). Initial and boundary conditions are reported in Table 4.

**Table 4.** Boundary conditions adopted for the injection molding simulation.

| Parameters | Values |
|---|---|
| Heated wall | Heat transfer coefficient 500 W (m$^2$ K)$^{-1}$<br>Temperature pulse 100 °C, 130 °C<br>Pulse duration 0 |
| Unheated wall | Heat transfer coefficient 2000 W (m$^2$ K)$^{-1}$<br>Temperature 20 °C |
| Inlet | Melt flow rate 4 cm$^3$ s$^{-1}$<br>Melt temperature 220 °C |
| Initial conditions | Air temperature 20 °C<br>Melt temperature 220 °C<br>$\phi_{air} = 1$<br>$\phi_{iPP} = 0$ |

## 5. Results

The heater was arranged below the cavity surface to modulate the cavity temperature during the process. Figure 3 shows pressure and temperature evolutions recorded in several positions along the flow path for the three injection molding conditions: conventional injection molding with constant and uniform mold temperature, and two heating powers applied to the heating device.

Pressure evolution in position P0 represents the pressure evolution in the nozzle: a pressure increase in the time range of −5–0 s due to the filling of the sprue and runner. At t = 0 s, the melt started to fill the cavity, the pressure abruptly increased, and a pressure peak was recorded when the cavity was completely filled. After the pressure peak, the packing stage began: pressure was kept at a constant value (200 bar) until the end of

packing (t = 3 s), when the pressure in the nozzle was released. Pressure evolution in positions P2 and P3 followed the same trend as pressure in position P0 up to the end of filling; after that, due to solidification, the pressure gradually decreased to zero.

Figure 3a shows the temperature evolution in positions P2 and P3 for the conventional case: the temperature was about 25 °C up to the first contact of the melt with the selected position; at this time the temperature achieved the maximum value. After the first melt–cavity contact, the temperature gradually decreased toward the value of the whole mold. At 80 °C and t = 5 s, a temperature inflection could be observed, corresponding to the incoming crystallization. Temperature evolution in position P3 was essentially similar to the temperature evolution in position P2, however, the values were generally smaller due to the cooling.

When the heater was active, a first temperature increase occurred due to the activation, before the melt's entrance into the cavity. The heating rate and the temperature that the cavity surface reached before the contact with the melt depend on the heating power: 68 W determined a higher temperature (180 °C) at the first melt–cavity contact than 40 W (150 °C). The temperature was kept high for an additional 0.5 s, which corresponds to the filling time; after that, it decreased toward the values of the whole mold. Pressure evolutions in positions P2 and P3 were not influenced by the heater activation, nor by the pressure peak or the pressure decrease rate after heater deactivation.

It is recognized that high temperatures during the filling stage promote the filling and consequently reduce the need for high filling pressures [25]. In the cases proposed in this paper, the temperature field was asymmetric: the temperature was modulated on one cavity side, whereas the other one was kept at the temperature of the whole mold. The asymmetric temperature distribution did not induce significant changes in the pressure evolutions, and thus the pressure necessary to fill the cavity was essentially the same as that required for the conventional case.

Figure 4 shows the optical micrographs of the specimens obtained in three injection molding conditions: conventional injection molding with a constant and uniform mold temperature, and two heating powers applied to the heating device.

Specimens were characterized by the oriented morphology typical of the injection molding process. In particular, the colored bands (blue and green) close to the sample surface represent the most oriented parts, namely the shear layer, usually characterized by structures developed along the flow direction [26]. The inner part of the sample, the brown area, is characterized by less oriented structures, spherulites in most cases [27].

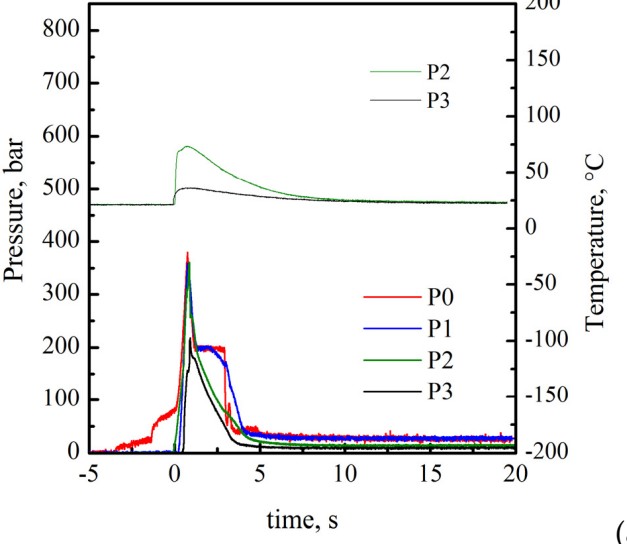

(a)

**Figure 3.** *Cont.*

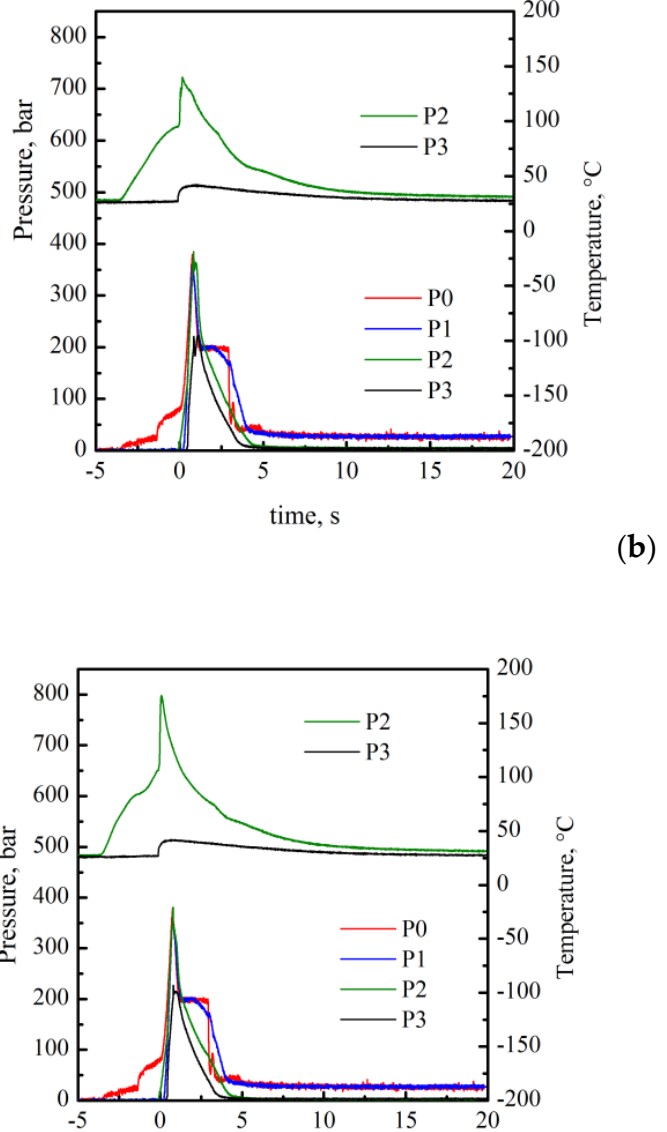

**Figure 3.** Pressure evolutions recorded in three different positions along the flow path, P0, P2, and P3, and temperature evolutions recorded in positions P2 and P3, where the heater is located, and position P3, where the heater is not present. Three cases are shown: (**a**) conventional injection molding condition, (**b**) 40 W heating power, and (**c**) 68 W heating power.

Figure 4a shows that the specimen obtained in conventional injection molding conditions was characterized by the widest oriented regions, covering the largest thickness. When the heating device was active (Figure 4b,c) on one side of the cavity, the oriented region strongly reduced. On the unheated side of the cavity, the oriented region extension was also affected by the heater activation on the opposite side. Figure 4d shows that the oriented region's thickness gradually decreased (to a smaller extent than on the heated side) when increasing the heating power.

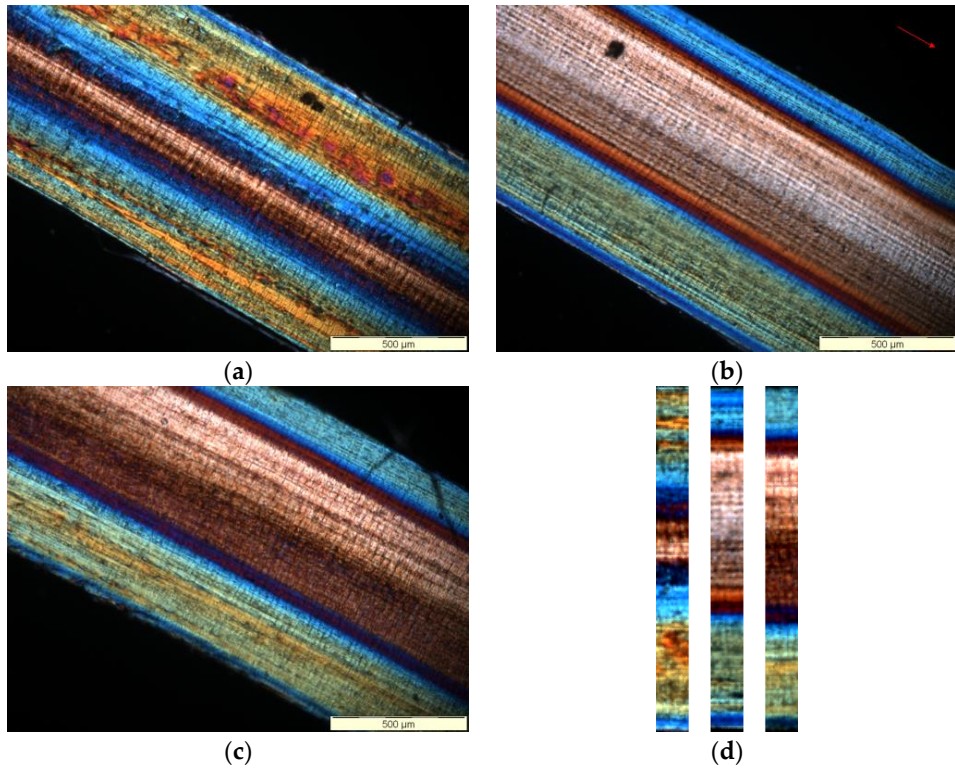

**Figure 4.** Optical micrographs acquired in position P2 for three cases: (**a**) conventional injection molding condition, (**b**) 40 W heating power, and (**c**) 68 W heating power. (**d**) Comparison among the three conditions.

Figure 5a shows that the morphology developed anticipating the activation time with respect to the polymer's entrance into the cavity. Figure 5b shows temperature evolutions recorded on the cavity surface at position P2. The heating duration after the first cavity–melt contact is comparable with the filling time (the same adopted for the cases shown in Figure 4).

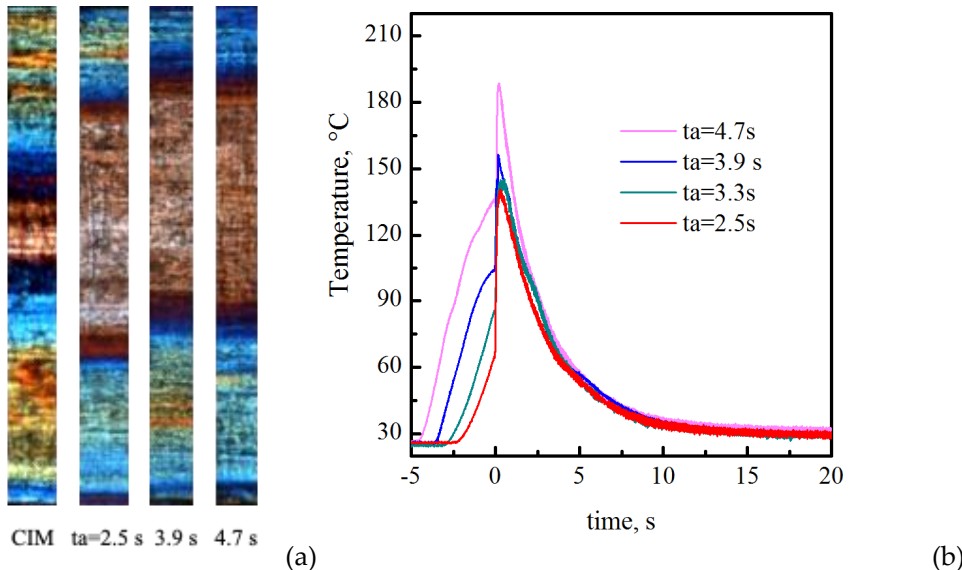

**Figure 5.** (**a**) Optical micrographs acquired at position P2 for four cases: conventional injection molding condition (CIM), and three activation times: 2.5, 3.9, and 4.7 s, with a 68 W heating power. (**b**) Temperature evolutions recorded at position P2 for the same cases shown in Figure 5a.

An earlier activation of the heater with respect to the polymer's entrance into the cavity allowed reaching a higher first-contact temperature (122 °C for ta = 2.5 s, and 188 °C for ta = 4.7 s). An increase of the activation time before the polymer's entrance into the cavity induced a reduction of the oriented layer on the heated side. The effect appears to be almost the same as that observed for the cases with modulation of the heating power. It can be stated that, when the heating time after the first melt–cavity contact is comparable with the filling time, the first-contact temperature determines the extension of the oriented layer.

## 6. Discussion

The observed results can be explained considering the melt behavior during the process in the presence of cavity temperature modulation. Process simulations allowed the analysis of melt behavior.

Simulation parameters, specifically the heat transfer coefficients adopted for the two cavity sides, were selected based on the materials adopted to construct the cavity insert. In particular, the higher heat transfer coefficient was selected based on [20], while the lower heat transfer coefficient, the one on the heated side, was selected accounting for the presence of several insulating layers. Model calibration was conducted based on the case with a 68 W heating power, where the purpose was to attain an accurate description of temperature evolution after heater deactivation on the heated side (see Figure 6).

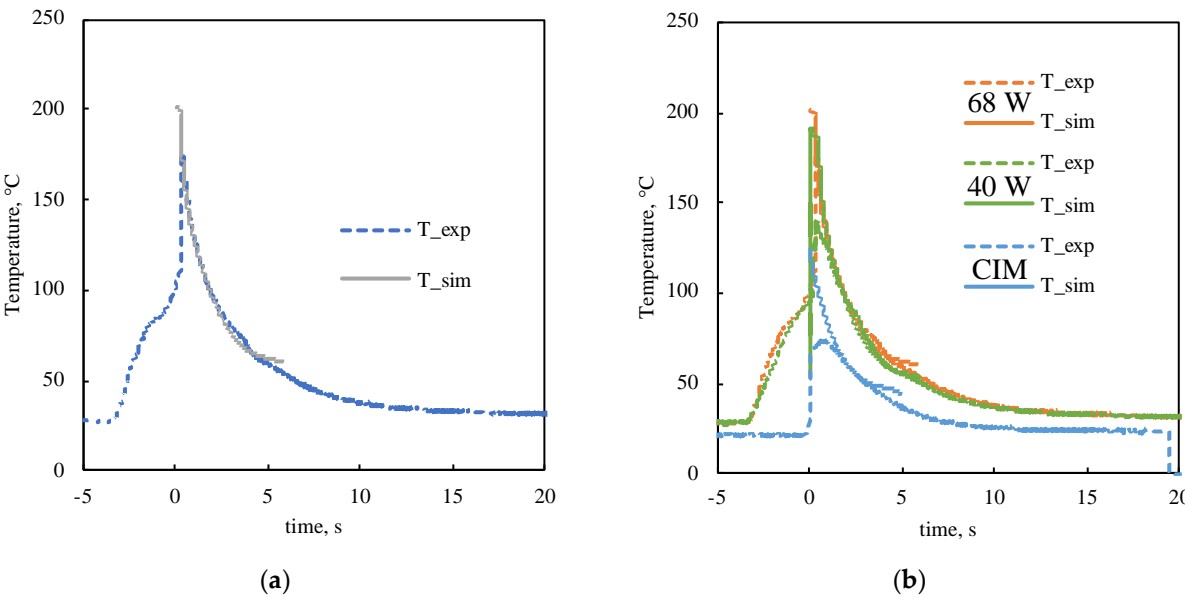

**Figure 6.** (**a**) Temperature evolution recorded on the cavity surface at position P2 for the case with a 68 W heating power. (**b**) Comparison with the simulated temperature evolution in the same position.

Figure 6a shows that the simulation consistently reproduced the temperature evolution; furthermore, temperature evolutions were consistently described for the other injection molding conditions, and this confirmed the selection of the proper heat transfer coefficient on the heated side.

Figure 7a shows the temperature distribution for the case with a 68 W heating power at several times during the process.

The temperature was high and almost constant along the cavity thickness during the filling (up to t = 0.5 s), and only the regions close to the cavity walls showed lower temperatures, depending on the presence of the heating device. In the time range of 0–0.05 s, the polymer front filled the whole thickness, and thus the temperature was high and equal to the melt temperature only in the cavity core. When the filling ended, the temperature decreased toward the mold values, even if some melt was still entering thanks to the packing stage. The temperature decrease was faster on the unheated side, whereas on the heated side, the temperature showed high values for a longer time, even after heater

deactivation. The uneven temperature distribution induced a non-symmetric filling of the cavity: the melt advanced faster on the heated side, as it can be observed in Figure 7b.

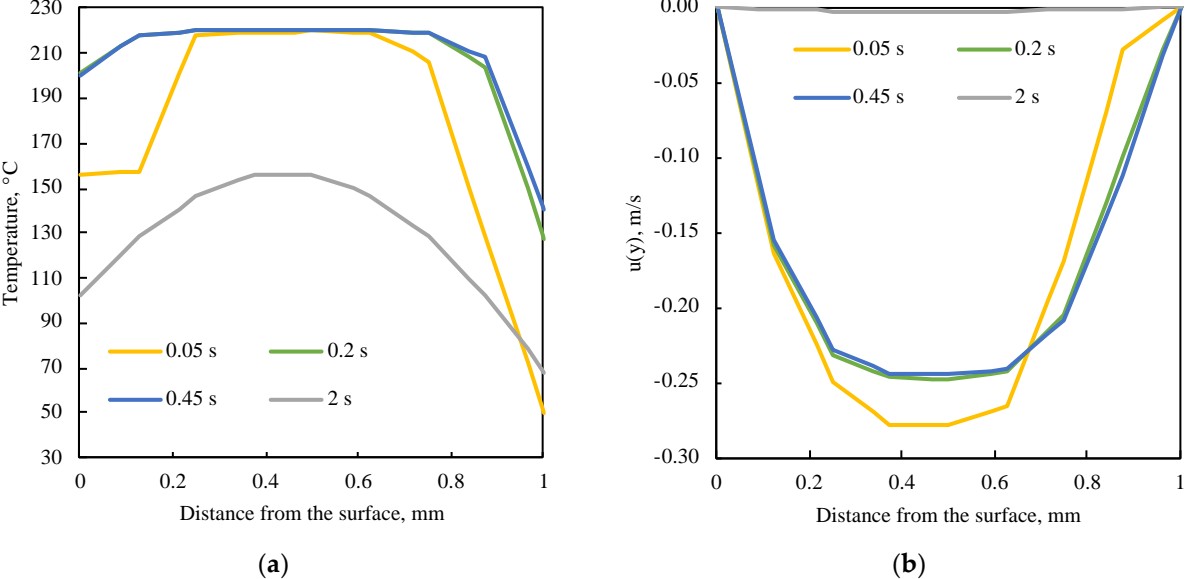

**(a)**                    **(b)**

**Figure 7.** (**a**) Temperature distribution along the cavity thickness for the case with a 68 W heating power. (**b**) Velocity distribution, y-component, along the cavity thickness at several times during the process.

The temperature and the flow field allow the evaluation of the molecular orientation in terms of Δ. Figure 8 shows the molecular orientation distribution for three cases: the conventional injection molding, 40 W, and 68 W heating power.

Figure 8a shows that the molecular orientation increased during the filling and never relaxed during the packing. Simulations confirmed that the molecular orientation was frozen during the packing due to the temperature decrease (inducing solidification), as it can be observed from Figure 8b referring to the case with a 68 W heating power at 0.214 mm from the heated side.

The conventional conditions showed an almost symmetric distribution of the molecular orientation, with two peaks in the shear layer (green band close to the cavity wall), small values in both the blue bands (one close to the cavity wall, the other one in the inner part toward the cavity core), and intermediate values in the core. When the heater was active, an asymmetric distribution of the molecular orientation was observed. In particular, a general decrease of the molecular orientation was observed in the regions close to the heated side, with the core showing the lowest values of orientation. This outcome is consistent with the decrease of the oriented region's thickness (green band disappeared and blue band reduced) close to the heated side. Additionally, the small values observed where the blue band is located moved toward the core (0.7 mm for the conventional condition, and 0.7 mm for the other two conditions). This finding is consistent with the decrease of the oriented region close to the unheated side (see Figure 3d).

The tools proposed for modulating the mold temperature jointly with the simulation code developed for predicting the molecular orientation distributions empowered the injection molding process opening the possibility to modulate part properties through the process.

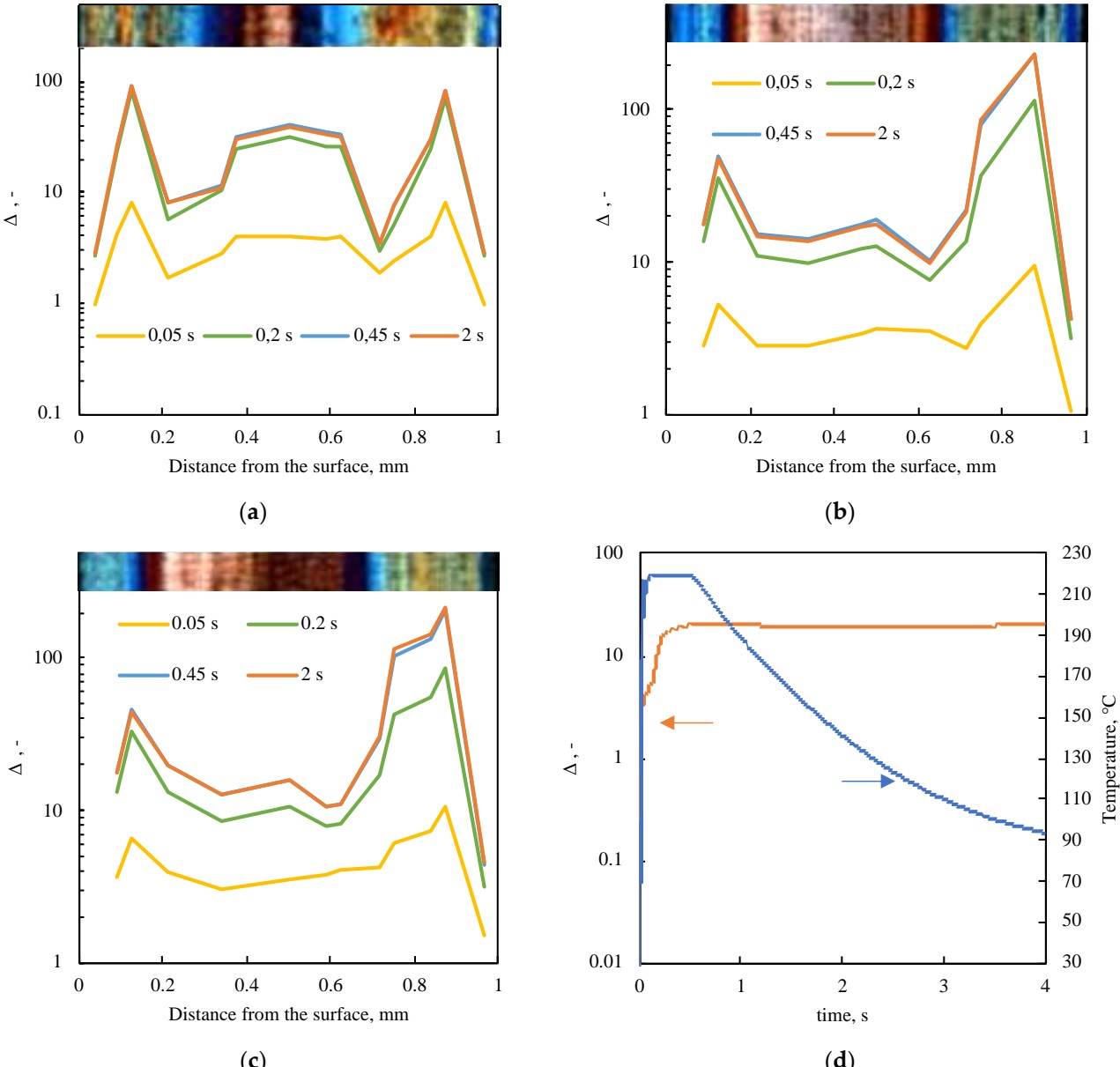

**Figure 8.** Temperature distribution along the cavity thickness for the (**a**) conventional injection molding case, (**b**) 40 W heating power, and (**c**) 68 W heating power. (**d**) Evolutions of temperature and molecular orientation for the case with a 68 W heating power at 0.214 mm from the heated side.

## 7. Conclusions

The uneven temperature distribution was analyzed during the injection molding process to assess its effect on the morphology developed in the mold. For this purpose, a thin electrical heater was designed and adapted below the cavity surface, allowing for the increase of the cavity surface temperature soon after the mold closure, and for a fast decrease of the mold temperature soon after the filling. The effect of several heating powers and heating durations on morphology was analyzed and exploited, considering the temperature and flow field realized during the process. In particular, the first melt–cavity contact temperature determined the number of oriented layers inside the mold: the oriented layer thickness decreased when increasing the temperature. The molecular orientation was calculated through process simulation conducted by Comsol. The asymmetric distribution of temperature along the cavity thickness induced an asymmetric filling of the cavity, and thus an asymmetric distribution of the molecular orientation during the process. The

molecular orientation obtained from the simulation was consistent with the experimental morphological observations.

**Author Contributions:** Conceptualization, S.L. and R.P.; methodology, S.L.; heating production, D.S.; software, S.L.; validation, S.L. and R.P.; formal analysis, R.P.; investigation, S.L. and D.S.; writing—original draft preparation, S.L.; writing—review and editing, S.L. and R.P.; supervision, R.P. All authors have read and agreed to the published version of the manuscript.

**Funding:** This research received no external funding.

**Institutional Review Board Statement:** Not applicable.

**Informed Consent Statement:** Not applicable.

**Data Availability Statement:** The data presented in this study are available upon request from the corresponding author.

**Conflicts of Interest:** The authors declare no conflict of interest.

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
