# Peer review of "Rapid Heating of Mold: Effect of Uneven Filling Temperature on Part Morphology and Molecular Orientation"

_processes, doi:10.3390/pr11010273_

Round 1
Reviewer 1 Report
This study proposed a strategy for controlling mold temperature during the process avoiding significant increase of the processing time. A thin electrical heater is design and adapted below the cavity surface, allowing for the increase of cavity surface temperature soon after the mold closure, and the fast decrease of mold temperature soon after the filling. The effect of several heating power and heating duration on morphology was experimentally analyzed and exploited considering the temperature and flow field realized during the process.
The topic is important to polymer industry community. The topic fits the scope of the journal Processes. The quality of the conducted study is moderate. The results of this manuscript are well presented and organized. The presented results of the performed modeling work have scientific and practical meaning. I appreciate your work on both simulations and experimental performance.
I almost have no comments and recommend to accept this manuscript after supplementation of the neglected details as follows.
1. Literature review should be improved. Maybe it would be better to highlight what you have discovered and what is the difference between your results obtained and those, which already available.
2. The description of models are to be improved. The solving method of the heat transfer equations may be given. Finite element method is ok, what’s more?
3. The authors mentioned the independency of numerical solution was carried out. What is the results?
4. It would be perfect if a validation against experiments is given.
Author Response
The authors thank the reviewers for the time and efforts devoted to revising the manuscript. We are confident that their suggestions significantly improved the manuscript's quality.
The answers to the reviewers' comments are detailed point-by-point. The manuscript was revised accordingly, and modifications have been highlighted in red.
Reviewer n.1
This study proposed a strategy for controlling mold temperature during the process avoiding significant increase of the processing time. A thin electrical heater is design and adapted below the cavity surface, allowing for the increase of cavity surface temperature soon after the mold closure, and the fast decrease of mold temperature soon after the filling. The effect of several heating power and heating duration on morphology was experimentally analyzed and exploited considering the temperature and flow field realized during the process.
The topic is important to polymer industry community. The topic fits the scope of the journal Processes. The quality of the conducted study is moderate. The results of this manuscript are well presented and organized. The presented results of the performed modeling work have scientific and practical meaning. I appreciate your work on both simulations and experimental performance.
I almost have no comments and recommend to accept this manuscript after supplementation of the neglected details as follows.
- Literature review should be improved. Maybe it would be better to highlight what you have discovered and what is the difference between your results obtained and those, which already available.
Literature review has been updated, and the difference between our results and the ones already reported in the literature has been clarified.
- The description of models are to be improved. The solving method of the heat transfer equations may be given. Finite element method is ok, what’s more?
The model description has been enlarged in the revised version of the manuscript.
- The authors mentioned the independency of numerical solution was carried out. What is the results?
The test for the independency of numerical solution on the mesh allows to find the proper mesh assuring reliable results and fast calculation time at the same time. This explanation is given in the revised version
- It would be perfect if a validation against experiments is given.
Numerical simulations were validated through comparison with the experimental temperature evolutions given in Figure 6b. The simulated temperature evolutions, reported as full lines, are consistent with the experimental one (dashed lines).
The revised manuscript is attached.

Reviewer 2 Report
This paper highlights controlling mold temperature during the injection molding process in order to assess its effect on the morphology developed in the molded. A systematic design of a thin electric heater has been design and adapted below the cavity surface to control the temperature and flow of material in mold cavity. A molecular orientation by means of process simulation conducted using Comsol has been calculated and correlated with the experimental result.
I like to thank the authors for crafting the manuscript. The manuscript is well written as far as English language is concerned. However, the manuscript contains high plagiarism. Detailed comments are given below. I request the authors to address the following comments for the improvement of the manuscript:
1. Plagiarism is high
2. Page 1, Line 8-9: Irrelevant sentence. It could be modified or deleted.
3. Page 1, Line 27-29: What is the relation between frozen polymer and length/thickness (L/T) ratio? This needs a detailed explanation and clarification.
4. Page 2, Line 46-48: Please explain each heating process and related problems which motivate the present investigation.
5. Page 2, Line 49-52: Avoid such a lengthy sentence.
6. Is there any adverse effect of temperature on the quality of the product using injection molding? Add a few pieces of information in the introduction section.
7. Page 2, Line 74-77: How do authors select the parameters used in the injection molding process?
8. Page 4, Line 135: What would be the effect of crystallization on the process? Provide a justification for not including the recrystallization effect.
9. Page 6, Line 184-186: Confusing statement regarding the three injection molding conditions. Arrange the condition as i), ii) and iii) ..
10. Figure 3: maintain a uniform sequence of data presentation, such as P0, P1..P3 instead of P2,P3,P0,P1 for all the sub-figures.
11. Use sentence style in all figure captions and embedded tests, For example, “Distance from the surface, mm” instead of “distance from the surface, mm” in Figure 7.
12. Author should highlight the crystallization at the interface of different temperature zones in the mold cavity. Does it lead to flow discontinuity?
Author Response
The authors thank the reviewers for the time and efforts devoted to revising the manuscript. We are confident that their suggestions significantly improved the manuscript's quality.
The answers to the reviewers' comments are detailed point-by-point. The manuscript was revised accordingly, and modifications have been highlighted in red.
Reviewer n.2
This paper highlights controlling mold temperature during the injection molding process in order to assess its effect on the morphology developed in the molded. A systematic design of a thin electric heater has been design and adapted below the cavity surface to control the temperature and flow of material in mold cavity. A molecular orientation by means of process simulation conducted using Comsol has been calculated and correlated with the experimental result.
I like to thank the authors for crafting the manuscript. The manuscript is well written as far as English language is concerned. However, the manuscript contains high plagiarism. Detailed comments are given below. I request the authors to address the following comments for the improvement of the manuscript:
- Plagiarism is high
Repetitions, which are mostly present in the materials and methods section, were reduced. In particular, the introduction was rewritten, also according to the reviewer n1 suggestions.
- Page 1, Line 8-9: Irrelevant sentence. It could be modified or deleted.
The sentence was deleted
- Page 1, Line 27-29: What is the relation between frozen polymer and length/thickness (L/T) ratio? This needs a detailed explanation and clarification.
The increase of the L/T ratio leads to a faster cooling, thus polymer solidification occurs more rapidly and, in some cases, the polymer solidifies before the cavity filling is complete. The concept has been more clearly expressed in the revised version of the manuscript
- Page 2, Line 46-48: Please explain each heating process and related problems which motivate the present investigation.
An explanation of the heating processes has been included in the revised version of the manuscript.
- Page 2, Line 49-52: Avoid such a lengthy sentence.
Long sentences have been removed
- Is there any adverse effect of temperature on the quality of the product using injection molding? Add a few pieces of information in theintroduction section.
In general, an increase of temperature is always beneficial for the quality of the product. Of course, this is true within the limit of thermal stability of the polymer and if the viscosity is high enough to avoid flash (material exiting the cavity). On the other hand, a higher temperature of the material also induces a longer cycle time. This is the reason why we adopted the thin heaters located on the surface, which allow a relatively fast cycle time.
- Page 2, Line 74-77: How do authors select the parameters used in the injection molding process?
A short sentence on the selection of injection molding parameters has been included in the method section.
- Page 4, Line 135: What would be the effect of crystallization on the process? Provide a justification for not including the recrystallization effect.
Morphology evolution was recognized to depend on the residual orientation achieved during the process; for this reason, the emphasis was given to the prediction of the residual molecular orientation in the molded. However, the incoming crystallization determine the start of solidification. This phenomenon was included through the introduction of a solidification temperature in the model describing the dependence of viscosity on the temperature.
- Page 6, Line 184-186: Confusing statement regarding the three injection molding conditions. Arrange the condition as i), ii) and iii) .
Obviously, the settled conditions were the same given in the method section, however, pressure and temperature in the cavity cannot be the same during the whole process, they change due to the material behaviour during the process. Figure 3 shows the recorded temperature and pressure in the cavity. The heating power adopted for the heating device affects the temperature achieved on the cavity surface before the melt contacts the cavity surface. A short explanation is included in the method section.
- Figure 3: maintain a uniform sequence of data presentation, such as P0, P1..P3 instead of P2,P3,P0,P1 for all the sub-figures.
Figure 3 has been revised
- Use sentence style in all figure captions and embedded tests, For example, “Distance from the surface, mm” instead of “distance from the surface, mm” in Figure 7.
Figure 7 has been updated
- Author should highlight the crystallization at the interface of different temperature zones in the mold cavity. Does it lead to flow discontinuity?
As already stated in the comment n.8, the effect of crystallinity was simplified by assuming a no-flow temperature. As the reviewer states, different temperature boundary conditions can cause different local evolutions of crystallinity and thus different solidification histories. This can give rise to a flow discontinuity in the sense that a volume downstream the solidified section cannot be reached by additional material. The model can handle this situation, which however never took place in the conditions adopted for this work. It should be considered that the crystallization kinetics of iPP generally increases on decreasing temperature reaching a maximum at about 50°C. This means that the heated sections always solidify at longer times, as also described by the no-flow temperature.
The revised version of the manuscript is included in the attachment

Round 2
Reviewer 2 Report
I thank the authors for modifying the manuscript as per the comments. The introduction section is substantially changed with a few additions of new citations.